# Morphological State Transition Dynamics in EGF-Induced Epithelial to Mesenchymal Transition

**DOI:** 10.3390/jcm8070911

**Published:** 2019-06-26

**Authors:** Vimalathithan Devaraj, Biplab Bose

**Affiliations:** Department of Biosciences and Bioengineering, Indian Institute of Technology Guwahati, Guwahati 781039, India

**Keywords:** epithelial to mesenchymal transition, morphology, phenotypic state transition, quantitative imaging, mathematical modeling, ultrasensitive switch, quasi-potential landscape

## Abstract

Epithelial to Mesenchymal Transition (EMT) is a multi-state process. Here, we investigated phenotypic state transition dynamics of Epidermal Growth Factor (EGF)-induced EMT in a breast cancer cell line MDA-MB-468. We have defined phenotypic states of these cells in terms of their morphologies and have shown that these cells have three distinct morphological states—cobble, spindle, and circular. The spindle and circular states are the migratory phenotypes. Using quantitative image analysis and mathematical modeling, we have deciphered state transition trajectories in different experimental conditions. This analysis shows that the phenotypic state transition during EGF-induced EMT in these cells is reversible, and depends upon the dose of EGF and level of phosphorylation of the EGF receptor (EGFR). The dominant reversible state transition trajectory in this system was cobble to circular to spindle to cobble. We have observed that there exists an ultrasensitive on/off switch involving phospho-EGFR that decides the transition of cells in and out of the circular state. In general, our observations can be explained by the conventional quasi-potential landscape model for phenotypic state transition. As an alternative to this model, we have proposed a simpler discretized energy-level model to explain the observed state transition dynamics.

## 1. Introduction

Epithelial to Mesenchymal Transition (EMT) is a phenomenon in which epithelial cells lose contact between neighboring cells and become semi-adherent, thereby acquiring migratory mesenchymal phenotype [1,2]. EMT is one of the possible mechanisms of cancer metastasis [3,4,5]. During EMT, cells switch between multiple phenotypes [6,7,8,9]. In general, change in the phenotype of a cell is considered as a transition of the cell from one state to another [10]. Cues from external signals [11,12] and the noise in the cellular system [13] can drive cellular state transition.

The metaphor of Waddington’s epigenetic landscape [14] is widely used to understand the directional state transition during differentiation [15]. In the generalized landscape model, cells move through a quasi-potential landscape with basins of attractions. The attractors are at lower potentials and are the preferred destination of cells. Each of those attractors is a particular phenotypic state [15].

The concept of potential landscape has been used to understand the phenotypic state transition dynamics of EMT [10,16]. Several authors have developed dynamical models of gene regulatory networks involved in EMT and created the potential landscape models based on those networks [17,18,19]. These studies have shown that the potential landscape of EMT has multiple attractors indicating that EMT is a multi-state transition process. 

Cellular state transition studies also help us to understand the lineages of different phenotypes. Gupta et al. [13] investigated phenotypic heterogeneity in a breast cancer cell line and had shown that stem-like cells emerge from non-stem-like cells through stochastic state transition. Su et al. [20] investigated the state transition dynamics in drug-induced resistance in melanoma. Using single-cell gene expression study, Mojtahedi et al. [12] showed that differentiation of progenitor cells to erythroid or myeloid lineage involves critical state transition. Hormoz et al. [21] used single-cell analysis to infer state transition paths in mouse embryonic stem cells and showed that these cells go through stochastic and reversible transitions along a linear chain of states.

Cellular states are commonly defined in terms of expression of molecular markers [13,20,22] or genome-wide expression profile [23]. Zhang et al. [9] have shown that TGF-β1-induced EMT in MCF10A cells involves transitions between three states defined by the relative expression of E-cadherin and Vimentin. Several other authors have also categorized phenotypic states during EMT in terms of expression of molecular markers and have developed mathematical models of multi-stable systems to explain the emergence of these phenotypes [6,7,8]. Here, the assumption is that the levels of expression of molecules reflect the phenotypic state of a cell. However, the state of a cell can also be defined by quantitative phenotypic features. For example, Kimmel et al. [24] used cell motility to define phenotypic states and investigated state transition behaviors in mouse cells.

In the present work, we have used the morphology of a cell to define its phenotypic state and investigated the dynamics of morphological state transition during Epidermal Growth Factor (EGF)-induced EMT. The key phenotypic signatures of EMT in cell culture-based models are the loss of cell-cell contact, change in morphology, scattering, and migration of cells [1,25]. These phenotypic features can be measured quantitatively and can be used to study phenotypic state transition [26,27,28]. 

We induced EMT in MDA-MB-468 cells using EGF. MDA-MB-468 is a triple-negative adenocarcinoma cell line of basal A type [29,30]. EGF-induced EMT of MDA-MB-468 cell is a well-established model for EMT [31,32,33,34,35,36]. We observed three distinct cell states based on the morphology of MDA-MB-468 cells. We call these cell states as cobblestone, spindle, and circular. We show that the spindle and the circular cells are the migratory cells. We have used quantitative image analysis to measure the population distributions of cells in these three states during EMT and estimated the state transition paths using a population dynamic model. Our model and estimation strategy can be used for any state transition system with aggregate data at discrete time points. We show that the state transition paths followed by MDA-MB-468 cells depend upon the dose of EGF and a critical state transition decision is controlled by an ultrasensitive on/off switch. As an alternative to the quasi-potential landscape model, we propose a discretized energy-level model to explain the observed state transition dynamics.

## 2. Methods

### 2.1. Cell Lines and Culture Conditions

Human breast cancer cell line MDA-MB-468 was procured from National Center for Cell Sciences, Pune, India and cultured in Dulbecco’s modified eagle medium (DMEM, Himedia, Mumbai, India) supplemented with 10% fetal bovine serum (Thermo Fisher Scientific, Waltham, MA, USA) at 37 °C in a humidified incubator with 5% CO_2_. For experiments with EGF (Shenandoah biotechnology 100-26, Warwrick, PA, USA) treatment, cells were maintained in reduced serum media (0.5% FBS in DMEM) for 12 h, followed by treatment in reduced serum media.

### 2.2. Phalloidin-FITC Staining

Cells were grown in 96 well plates. After EGF treatment, cells were fixed with 4% paraformaldehyde for 10 min at room temperature. Cell membrane was permeabilized using 0.1% Triton X-100 in PBS, and the cells were stained with 0.1 µM of FITC Phalloidin conjugate in PBS for 1 h. Cells were counterstained with 30 µM DAPI in PBS for 5 min at room temperature. Cells were washed twice with PBS followed by imaging using an Epi-fluorescence microscope (Nikon Eclipse Ti-U, Nikon Instruments Europe BV, Amsterdam, The Netherlands).

### 2.3. Immunofluorescence

Cells were grown in 12 well glass chamber slide (Ibidi 81201, Grafelfing, Germany). After treatment, cells were fixed with ice-cold methanol and acetone in 1:1 ratio at −20 °C for 10 min. Cells were incubated with permeabilization buffer for 10 min at room temperature, followed by incubation with blocking buffer (1% BSA, 0.3 M glycine in PBS containing 0.1% Tween20) for 30 min at room temperature. Cells were stained overnight with fluorophore-conjugated primary antibody at 4 °C. Cells were washed twice in PBS followed by imaging using a confocal microscope (Zeiss LSM 880, Carl Zeiss Microscopy, LLC, Thornwood, NY, USA). Details of the antibodies used are given in Appendix A.

### 2.4. Quantitative PCR

RNA was isolated using TRI reagent (Sigma, St. Louis, MO, USA) followed by Turbo DNAse (Thermo Fisher Scientific, Waltham, MA, USA) treatment to get rid of genomic DNA contamination. RNA was reverse transcribed using Verso cDNA synthesis kit (Thermo Fisher Scientific, Waltham, MA, USA). qPCR was performed using Quantifast SYBR Green (Rotor-Gene Q, QIAGEN, Hilden, Germany). All experiments were done in triplicates and normalized to cyclophilin A. Data analysis was done using LinRegPCR [37]. Primers used in qPCR are listed in Appendix A.

### 2.5. Quantitative Image Analysis

Cells were grown in 96 well plates. After treatment, cells were fixed with 4% paraformaldehyde for 10 min at room temperature. Cell membrane was permeabilized using 0.1% Triton X-100 in PBS, and the cells were stained with HCS cell mask red dye (Thermo Fisher Scientific, Waltham, MA, USA) at a final concentration of 0.001 µg/µL for 1 h. Cells were imaged using Epi-fluorescence microscope (Nikon Eclipse Ti-U, Nikon Instruments Europe BV, Amsterdam, The Netherlands). Ten non-overlapping fields of view were taken for each experimental condition. Image segmentation, object identification, and extraction of geometric features were accomplished using CellProfiler [38]. Classification of cells was carried out using CellProfiler Analyst through machine learning algorithm provided with it [39]. A set of rules was generated by training the tool with images that contain all possible cell types. Using those rules the experimental images were classified. Appendix A shows the quality of the training and the accuracy of the predictions from the trained data set.

### 2.6. Migration Assay

Cells were grown in transwell inserts in 24 well plates (Polycarbonate cell culture inserts with 8-micron pore size, Thermo Fisher Scientific, Waltham, MA, USA) with reduced serum media in both the insert as well as the plate. After 24 h of EGF treatment, transwell inserts were placed in a fresh plate with reduced serum media. After 6 h, cells lying within the insert were removed using a cotton swab and cells that had migrated to the other side of the membrane were fixed with 100% ice-cold methanol. Cells were stained with HCS cell mask red dye (Thermo Fisher Scientific, Waltham, MA, USA), and the migrated cells were imaged using a confocal microscope (Zeiss LSM 880, Carl Zeiss Microscopy, LLC, Thornwood, NY, USA).

### 2.7. Western Blotting

Cells were grown in 35 mm dishes. MDA-MB-468 cells were treated with different doses of EGF for different time points as mentioned in the results section. Whenever required, cells were treated with pathway inhibitor Gefitinib (Abcam ab142052, Cambridge, UK). After treatment, cells were lysed in RIPA buffer containing PMSF (1 mM), sodium orthovanadate (1 mM), sodium fluoride (50 mM) and EDTA (1 mM). Total protein was estimated by Lowry’s method [40]. An equal amount of lysate from each sample was resolved by SDS PAGE and transferred to PVDF membrane by wet transfer. The membrane was blocked by 3% BSA in TBST for 2 h, followed by overnight incubation with primary antibody at 4 °C. Target proteins were detected by chemiluminescence (SuperSignal West Dura kit, Thermo Fisher Scientific, Waltham, MA, USA) using HRP conjugated secondary antibody. Developed blots were imaged using a gel documentation system (ChemiDoc XRS+, BioRad, Hercules, CA, USA). Detected bands were quantified by densitometry using ImageJ [41]. Target proteins were normalized with respect to loading control. Details on the antibodies used are provided in Appendix A.

### 2.8. Flow Cytometry

Cells grown in 35 mm dishes were treated with different doses of EGF for different time points as mentioned in the results section. Cells were trypsinized and re-suspended in PBS followed by methanol fixation (final concentration of 80% methanol). Fixed cells were kept in −20 °C for 15 min. Cells were incubated with blocking buffer (0.5% FBS in PBS) for 2 h at room temperature. The cells were stained overnight with primary antibody at 4 °C. Subsequently, the cells were stained with the Alexa Fluor 488 conjugated secondary antibody and analyzed in CytoFLEX (Beckman Coulter, Brea, CA, USA). The positive population was estimated by Overton histogram subtraction. Cells stained with only secondary antibody was used as a control in histogram subtraction. Data analysis was done using FCS Express 5 (De Novo Software, Glendale, CA, USA). Details of the antibodies used are given in Appendix A.

### 2.9. Live and Dead Cell Estimation

We have used the method developed by Dengler et al. [42] and Wan et al. [43]. Cells were grown in 96 well plates. After treatment, propidium iodide (PI) was added at a final concentration of 1 µg/mL into each well without removing the media. Cells were incubated at 37 °C for 10 min. Dead cells with compromised membrane would take up PI. Fluorescence was measured using a microplate reader (Infinite M200 PRO, Tecan, Mannerdorf, Switzerland) at λ_ex_ = 530 nm and λ_em_ = 620 nm. Fold change in dead cells was estimated with respect to time *t =* 0 sample. Similarly, the change in the total cell population was also estimated. After treatment, staining solution (final concentration: 30 µg/mL of PI, 0.1 M EDTA, 0.5% Triton X-100) was added into each well without removing the media. Cells were incubated for 6 h at room temperature followed by fluorescence measurement. Percentage live and dead cells were estimated from this data. A standard curve was plotted to check the linear regime of the assay (Appendix A).

### 2.10. Cell Viability Assay

MDA-MB-468 cells were seeded in 96 well plates. Cells were treated with different doses of Gefitinib for different time points. Subsequently, the viability of the cells was measured by 3-(4,5-dimethylthiazol-2yl)-2,5-diphenyltetrazolium bromide (MTT) assay [44]. DMSO was used as a solvent for Gefitinib. The percentage of cell viability was calculated relative to cells treated with an equivalent amount of DMSO in media (without Gefitinib).

### 2.11. Mathematical Model

A state transition model was developed to understand the dynamics in EGF-induced cell state transition. Experimental observations of cell state distribution and fold change in total cell population upon EGF treatment were used as input to the model. From the model we estimated the fraction of cells moving from one state to another state in a particular time interval. Details of the model and the estimation procedure are given in the Appendix A. Parameter estimation and analysis of the model were done using MATLAB 2018a. The estimated parameters are given in Appendix A.

### 2.12. Data Analysis

SigmaPlot was used to generate graphs and for statistical analyses. Mean of multiple data points are plotted with error bars representing standard deviations. Wherever applicable, suitable statistical tests were performed and are mentioned in respective figure legends/text.

## 3. Results

### 3.1. EGF-Induced EMT

We treated MDA-MB-468 cells with different doses of EGF to induce EMT. Cells were stained with Phalloidin to visualize the change in F-actin distribution and cell morphology. MDA-MB-468 cells grow as a monolayer of cobblestone-shaped cells attached to each other. Upon EGF treatment, the morphology of these cells changed, and they lost cell-cell contacts (Figure 1a). 

Quantitative PCR showed that EGF-treated cells had higher expression of Vimentin, Fibronectin, Snail1, and Zeb1 (Figure 1b). Immunofluorescence imaging confirmed the increased expression of Vimentin and Snail1 post-EGF-treatment (Figure 1c). Our observations in changes in morphology and gene expression are in accordance with earlier reports of EGF-induced EMT in MDA-MB-468 cells [32,34,45].

### 3.2. Morphological States of MDA-MB-468 Cells

Cells were stained with HCS cell mask red dye and imaged using a fluorescence microscope to observe EGF-dependent change in morphology (Figure 2a). We observed that in our experimental system, MDA-MB-468 cells had three distinct morphologies. We call these cells cobblestone, spindle, and circular cells (Figure 2b). Cobblestone cells were polygonal with cell-to-cell contact. Spindle cells and circular cells were scattered and loosely adhered. All these three cell types were in monolayer, and none of them were floating over the medium.

Through image analysis, we estimated the percentage of each cell types in a population. It was observed that the population distribution of these cells changed with the dose of EGF (Figure 2c). We considered these three morphologies as three phenotypic states.

### 3.3. Functional Characterization of Three Cell States

We have done experiments to categorize the cell types based on their physiological functions. Through image analysis, we quantified the extent of scattering of cells upon EGF treatment. For each cell, we measured the number of nearest neighbors. For scattered cells, the number of nearest neighbors would be lesser than that of cells in a cluster. As shown in Appendix A, the number of nearest neighbors for circular and spindle cells were lesser than that of cobble cells. The circular cells were found to be more scattered than spindle cells. The median number of nearest neighbors for the spindle and circular cells were one and zero, respectively.

We checked the migratory potential of these cells using the Boyden Chamber assay (Appendix A). In the absence of EGF, very few cells migrated to the other side of the membrane, and they were spindle and circular. This shows that spindle and circular cells are inherently migratory phenotypes. We performed the same experiment in the presence of different doses of EGF. As shown earlier (Figure 2c), EGF treatment favored the formation of circular and spindle cells. In the Boyden Chamber assay for EGF-treated cells, a large number of cells migrated to the other side and they were again circular and spindle types. Therefore, circular and spindle cell states are the migratory phenotypes, while cobble cell state is a non-migratory phenotype.

Franchi et al. [46] have earlier shown that membrane filters used as inserts in cell culture experiments affects the morphology of MCF-7 cells. However, we did not observe such an effect of membrane insert on the morphology of the cells in our Boyden chamber assay.

### 3.4. Dose-Dependent Temporal Dynamics of State Transition

The time-dependent changes in the distribution of cells in three morphological states for different doses of EGF are shown in Figure 3. The population of cells remained in a steady state distribution in the absence of EGF (Figure 3a). The steady state distribution had the majority of cobble cells (79% ± 4%) and a minor proportion of spindle (13% ± 2%) and circular (8% ± 2%) cells.

At moderate doses of EGF (5 and 10 ng/mL), we observed reversible population dynamics with an initial rise in circular cells, followed by an increase in spindle cells and eventually the population distribution returned towards the initial state (Figure 3c,d). However, at a lower dose of EGF (1 ng/mL), a marginal increase in the spindle cell population was observed, but changes in population distribution were not statistically significant (ANOVA, *p >* 0.01). At a high dose (25 ng/mL EGF), cells mostly remained in the circular state till 60 h (Figure 3e).

### 3.5. Trajectories of Cell State Transition

The population dynamics observed in our experiments can emerge when cells jump from one phenotypic state to another depending upon the external cue. We used a mathematical model to estimate the state transition trajectories from the imaging data. Usually, cell state transition models are time-homogenous steady-state models that do not consider the death and birth of cells [13,20,47]. However, we have observed that our experimental system was not conserved and there was a change in the total number of cells with time (Appendix A). Further, reversible change in population distribution observed in our experiments ruled out the assumption of steady-state and time homogeneity.

We created a discrete-time population dynamics model that considers all possible state transitions along with birth and death of cells (Appendix A). This is a generic model that can be used for any experimental system where aggregate population data is collected at discrete time intervals. We estimated the cell state transition parameters for each time interval by fitting the model to image analysis data (Appendix A).

Figure 4a shows the state transition diagram for cells treated with 10 ng/mL of EGF. From the estimated state transition parameters, we have calculated the normalized flux of cells through each path at each time intervals. Normalized flux represents the fraction of live cells moving through a particular path in a particular time interval. As shown in Figure 4a, the main flux of cells was in the cobble → circular → cobble path (solid black arrow).

For cells treated with 10 ng/mL of EGF, a substantial increase in the population of spindle cells was observed at 36 h (Figure 3d). The state transition model shows that this increase was due to the transition of some cells from the circular state to the spindle state in the interval of 24 h to 36 h (Figure 4a, blue line) and contributions of other state transition paths were minor.

The 24 to 36 h time interval is crucial as two branching processes were observed in this interval—circular → cobblestone and circular → spindle. To further investigate, we had additional observations at three hour intervals in this period (Appendix A).

The state transition diagram for this expanded time interval is shown in Figure 4b. At 24 h, the majority of cells were in the circular state. At subsequent time intervals, a portion of these cells moved to spindle state. These cells in the spindle state followed two paths - either they stayed in the same state or moved to cobblestone state. Therefore, the reversal from circular to cobblestone state had a transition through the spindle state.

The key inferences from this state transition analysis are—(a) cell state transition in EGF-induced EMT of MDA-MB-468 cells is reversible, (b) the dominant state transition path for cells treated with a moderate dose of EGF is cobble → circular → spindle → cobble, and (c) spindle cells are predominantly formed from circular cells. Therefore, the emergence of spindle cells requires the transition of cells from cobblestone to circular state.

We also constructed the state transition diagram for cells treated with 25 ng/mL of EGF (Appendix A). At this high dose of EGF, the dominant state transition path was cobble → circular. Since there was no reversal to cobble state, we did not observe any substantial increase in the spindle cell population in this experiment. 

Our state transition model considered all possible paths of state transition along with cell death and birth. However, it can be hypothesized that the observed changes in the population distribution were due to preferential death and birth of cells in specific states and there was no cell state transition. To check the validity of this alternative hypothesis, we created a null model that consider birth and death of cells, but do not consider any cell state transition (Appendix A). 

The estimates from this model for the change in cell number and the extent of cell death did not match with our experimental observations (Appendix A). Instead, this model predicted very high and unrealistic cell death (Appendix A). Therefore, we rejected the null model.

### 3.6. Dynamics of EGF Signaling Drives the State Transition

The signal given by EGF gets encoded first in the temporal dynamics of phosphorylation of its receptor EGFR. In MDA-MB-468 cells, EGF induced a transient change in phosphorylation of EGFR, with a very fast rise followed by a gradual decline (Figure 5a). It was observed that the level of EGFR phosphorylation and the rate of its decay depend upon the dose of EGF.

We also used flow cytometry to detect phosphorylation of EGFR. The temporal dynamics of EGFR phosphorylation observed in this experiment was similar to the results of Western blot experiments (Appendix A). In all cases, the distributions of cells were broad but unimodal, indicating the absence of any distinct subpopulation. With EGF treatment, the whole population of cells moved to a higher level of EGFR phosphorylation and then with time, shifted back to a lower level.

In our experiments, we have observed that the circular and spindle cells were scattered and migratory. Lu et al. [48] have shown that EGF signaling promotes cell invasion and metastasis through dephosphorylation of Focal Adhesion Kinase (FAK), a key molecule involved in cell adhesion to the extracellular matrix [49,50,51]. In untreated MDA-MB-468 cells, phosphorylation of FAK was high (Appendix A). In EGF-treated cells, phospho-FAK declined and returned to its high level only when the phospho-EGFR level dropped (compare Appendix A with Figure 5a). Therefore, the temporal dynamics of phospho-FAK was correlated with the temporal dynamics of phospho-EGFR and the population dynamics of cells.

From the observations of phospho-EGFR dynamics and the population distribution data, we can hypothesize that the cells move to circular state only when phospho-EGFR shoots very high. On the other hand, the transition to spindle cell state happens in the decay phase of phospho-EGFR dynamics. Subsequently, most of the cells return to the cobblestone state when phospho-EGFR reaches the basal level. In case of prolonged phospho-EGFR activation (like in 25 ng/mL EGF treated cells), cells remain stuck in the circular state for a longer duration.

To further understand the relationship between the dynamics of phospho-EGFR and cell state transition, we treated cells with two pulses of 10 ng/mL of EGF (at *t =* 0 and *t =* 12 h). In comparison to one dose of EGF, two pulses of EGF generated a higher phospho-EGFR level for a longer duration (Figure 5b). Similarly, when two pulses of EGF were given, most of the cells stayed in the circular state for a much longer duration than one dose of EGF (Figure 5c). These observations strengthened our hypothesis that high phospho-EGFR level is required to move cells to circular state and to keep those in that state.

### 3.7. An Ultrasensitive Switch-Like Response in State Transition

To further substantiate the hypothesis that the level of phospho-EGFR controls the cell state transition, we have plotted proportion of the circular cells against the level of phospho-EGFR in different experimental conditions (Figure 6). This figure resembles an ultrasensitive on/off switch wherein a small change in the input signal triggers a drastic change in the response [52]. The grey shaded region in Figure 6 represents the ultrasensitive region, where a slight shift in phosphorylation of EGFR will have a large impact on the circular cell population.

We perturbed this on/off switch using Gefitinib, an EGFR inhibitor. First, we treated cells with a high dose of EGF (25 ng/mL), which induced the phosphorylation of EGFR, thereby turning the switch ON. After 12 h, we added Gefitinib (0.2 µM) and turned the switch OFF. The dose of Gefitinib was much below its IC_50_ value (Appendix A).

On EGF treatment, most cells initially became circular (Figure 7c). However, when Gefitinib was added at 12 h, cells started to revert from the circular state to the spindle and cobblestone state (Figure 7a). Additionally, phosphorylation of EGFR dropped, and Phospho-FAK increased immediately after Gefitinib treatment (Figure 7a). These observations confirmed that there exists an ON/OFF switch involving phospho-EGFR that decides whether a cell will be in the circular state or not.

## 4. Discussion

Epithelial to mesenchymal transition involves the transition of cells through multiple phenotypic states. In this work, we have identified state transition trajectories in an in vitro model system for EMT using quantitative imaging and mathematical modeling. We used EGF to induce EMT in our experiments and have elucidated the link between the temporal dynamics of EGFR phosphorylation and the cellular state transition dynamics.

Phenotypic state transition happens through two mechanisms—by stochastic fluctuation and by the instruction of an external cue [12,54,55]. Both of these are understood in terms of a quasi-potential landscape with multiple attractors representing distinct phenotypic states. In the first mechanism, cells move from one phenotypic state to another due to stochastic fluctuation [13]. This leads to a steady state distribution of cells in different phenotypic states. In our work, we have categorized cells in three morphological states. In untreated condition, the relative proportions cells in these three states remained almost constant throughout our observations. 

An external cue changes the potential landscape pushing cells from one state to another [12,56]. This gives directionality to state transition and deviates the population distribution way from the steady-state distribution. For a time-varying input signal, the changes in the landscape will depend both on the strength of the signal and time. 

In our experiments, we have observed both of these time- and dose-dependent effects. For moderate doses of EGF, activation of EGFR was short, leading to a reversible population dynamics. On the other hand, a high dose of EGF caused prolonged activation of EGFR and cells followed the unidirectional path cobble → circular. 

We have observed that the spindle cells emerged primarily from the circular ones during the decay phase of EGFR phosphorylation. During this phase, the dominant course of state transition was circular → spindle → cobble; whereas, during the activation phase of EGFR, the path was cobble → circular. This means that the changes in the quasi-potential landscape during EGFR activation phase is different from that in the decay phase. The idea of signal-dependent change in the quasi-potential landscape and associated population dynamics is further substantiated in our experiments where two pulses of EGF were given, and cells were treated with an EGFR-inhibitor. 

We have observed that transitions in and out of the circular state are linked with the phosphorylation status of EGFR and the relation is ultrasensitive. An ultrasensitive switch helps a cell in making all or none decision [57,58]. One of the canonical pathways activated by EGF is the MAPK pathway. MAPK pathway is known to have ultrasensitive switch-like behavior [59]. Melen et al. [60] have shown that during embryonic development of Drosophila cell state change triggered through EGFR activation has ultrasensitive behavior.

As discussed above, our observations on the state transition dynamics in EGF-induced EMT in MDA-MB-468 cells are in accordance with the concept of the quasi-potential landscape for phenotypic states. However, our observations can be explained by another formulation of state transition. In this, let us consider the phenotypic states as discrete, and each of those corresponds to a discrete energy level. We have observed that, in the absence of EGF, cells had at an apparent steady-state distribution of states– Cobble: Spindle: Circular = 0.79:0.13:0.08. In quasi-potential models, the steady state probability of a cell being in particular state is linked to the potential of that state according to the relation U=−ln(p), where *U* and *p* are the dimensionless potential and steady state probability respectively [15,18,61]. Therefore, we can calculate the potential for each of the cellular states in our experimental system as $U_{i}=-\ln\left(f_{i}\right)$, where *f_i_* is the fraction of cells in the *i^th^* state at steady state, *i = cobble, spindle, circular* [62]. This allows us to draw a state diagram with phenotypic states arranged vertically as per their potentials (blue horizontal lines in Figure 8). This diagram is similar to the Jablonski diagram [63] used to represent state transition in molecular spectroscopy.

EGF treatment causes rapid activation of EGFR. During this fast activation, cells move to the circular state that has the highest potential (green arrow Figure 8). This is equivalent to transition from the ground state to an excited state in the Jablonski diagram. On the other hand, the decay of phospho-EGFR is a slow process, and the relaxation in the energy level of a cell is also slow. Therefore, a cell at the circular state does not jump directly to the lowest potential, but first jumps to spindle state that has the second highest potential (red arrows Figure 8).

All state transitions are still probabilistic. The dose of EGF decides the probability of transition of a cell from the lowest potential to the highest one and the rate of relaxation. For a low dose of EGF, cells fail to move to the circular state; whereas a moderate to high dose of EGF forces most of the cells to the circular state. Similarly, the dose of EGF controls the rate of decay of phospho-EGFR and in turn, controls the speed of relaxation of cells from circular state to the lower potential states. Therefore, within 60 h, we have observed the circular → spindle → cobble transition in cells treated with 10 ng/mL of EGF, but have not observed the same for 25 ng/mL EGF treatment. 

This formulation of discrete energy states has certain advantages over the conventional potential landscape model. Cell-based experiments like those in this paper never give complete empirical information about the shape of the potential landscape. Therefore, understanding the changes in the potential landscape with time and external cues is difficult. Generating a potential landscape is difficult when discrete states are defined using structural or functional aspects of a cell rather than the expression of specific markers. Our formulation does not have these limitations. As shown, one can calculate the potentials corresponding to discrete phenotypic states from steady-state data and then draw the state diagram that can be used to explain state transition trajectories. Further, this model is much simpler than the landscape model and readily amenable to stochastic modeling to explain the state transition behaviors.

Most of the previous studies on state transition in EMT were focused on specific molecular networks. Accordingly, cellular states were defined in terms of expression of specific molecular markers. Even though EMT is a complex process involving a large number of molecules, such studies on specific networks involving a handful of molecules are essential for understanding EMT. However, irrespective of our focus on a particular molecular network, EMT involves specific phenotypic changes in cells. These phenotypic changes can be measured quantitatively, and the measurement techniques can be scaled up for high-throughput experiments. In our work, we have used morphologies of cells to define phenotypic states. We have shown that quantitative image analysis can be used to study the dynamics of morphological state transition in an in vitro experiment. One can implement a similar strategy to study EMT in terms of other phenotypic features of EMT like migration potential of cells. Such studies involving direct characterization of quantitative phenotypic traits would augment molecular marker-based investigations.

## Figures and Tables

**Figure 1 jcm-08-00911-f001:**
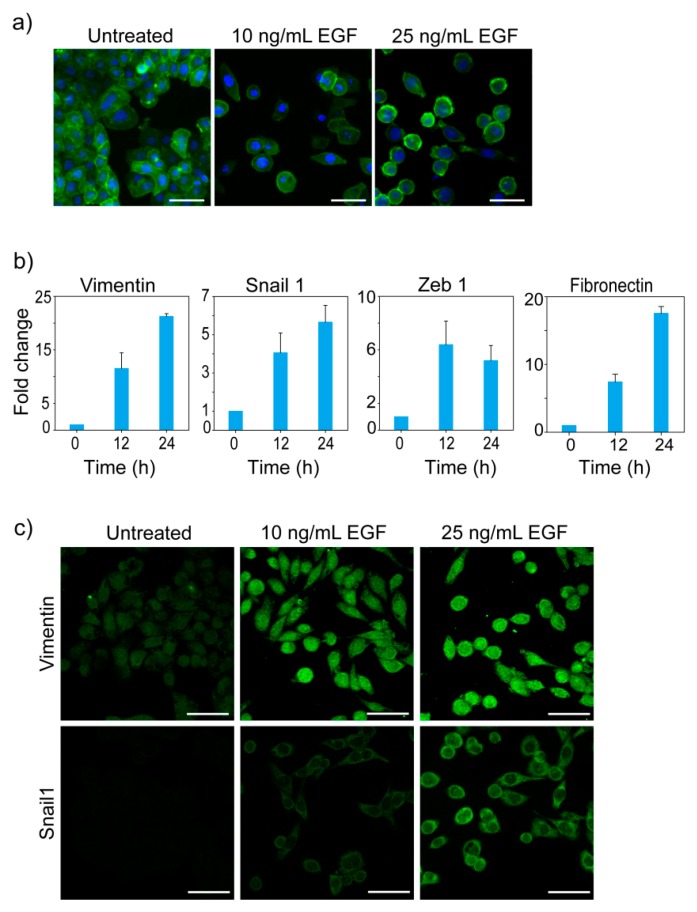
EGF induces EMT in MDA-MB-468 cells. (**a**) Cytoskeletal reorganization and change in morphology. After 24 h treatment with different doses of EGF, cells were stained with Phalloidin and DAPI. Green and blue colors represent the cytoskeleton and DNA content respectively. (**b**) Expression profile of EMT related genes. Cells were treated with 10 ng/mL of EGF and the fold change in expression was measured by qPCR. Averages of three measurements are shown with error bar representing standard deviation. Observed changes in expression of all the genes were statistically significant (Kruskal-Wallis analysis of variance, *p <* 0.01). (**c**) Immunofluorescence imaging of Vimentin and Snail1. Cells were treated with different doses of EGF for 24 h and stained with Fluorescent-dye conjugated anti-Vimentin and anti-Snail1 antibodies. Scale bar in images: 50 µm.

**Figure 2 jcm-08-00911-f002:**
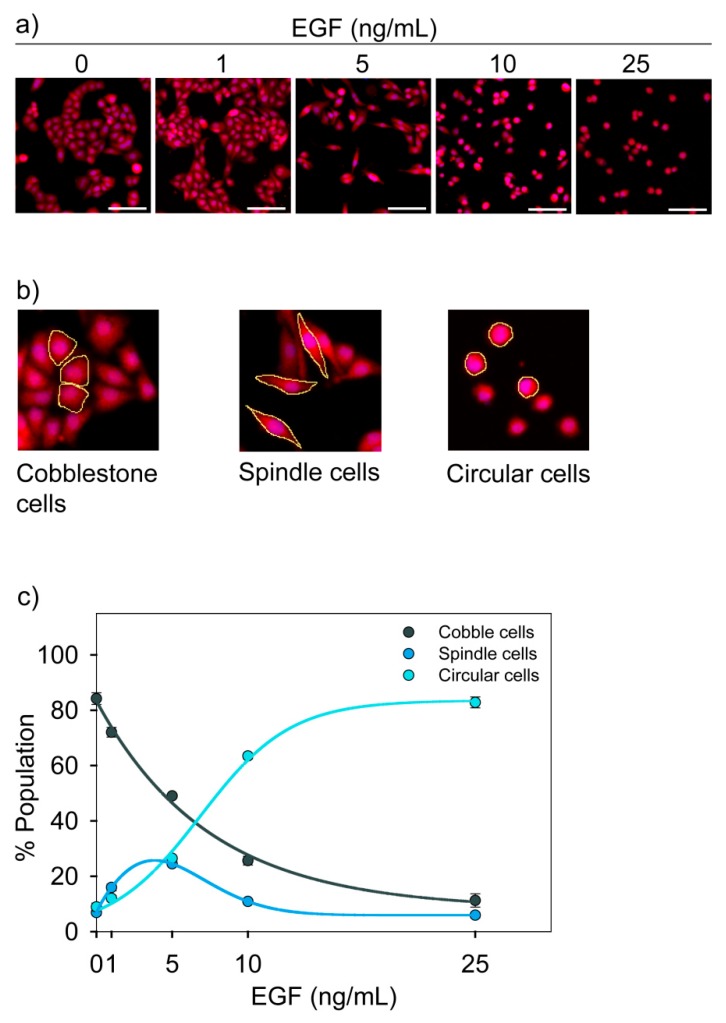
EGF-induced change in morphology of MDA-MB-468 cells. Cells were treated with different doses of EGF for 24 h, and the change in morphology was imaged by fluorescence microscopy. (**a**) Representative images for each dose of EGF show the dose-dependent effect of EGF on the morphology. Scale bar: 100 µm. (**b**) Cells with three distinct morphologies were observed. These are named as cobblestone, spindle and circular. Typical cell types in each category are highlighted by the yellow line. (**c**) EGF-induced change in population distribution. The graph represents quantitative data from image analysis. Each data point represents the mean of three independent experiments and error bars indicate standard deviation.

**Figure 3 jcm-08-00911-f003:**
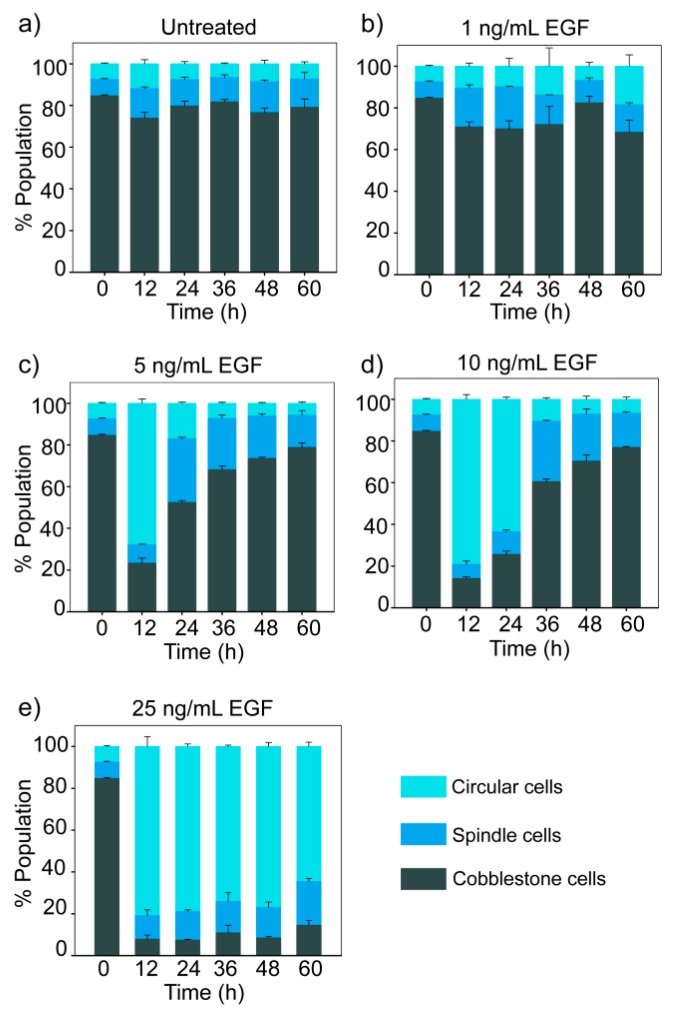
Dose- and time-dependent effect of EGF on population distribution. MDA-MB-468 cells were treated with different doses of EGF for different durations, and cells were imaged by fluorescence microscopy. The graph represents quantitative data from image analysis. Each data point represents the mean of three independent experiments and error bars indicate standard deviation. (**a**) In the absence of EGF; (**b**) 1 ng/mL EGF; (**c**) 5 ng/mL EGF; (**d**) 10 ng/mL EGF; (**e**) 25 ng/mL EGF.

**Figure 4 jcm-08-00911-f004:**
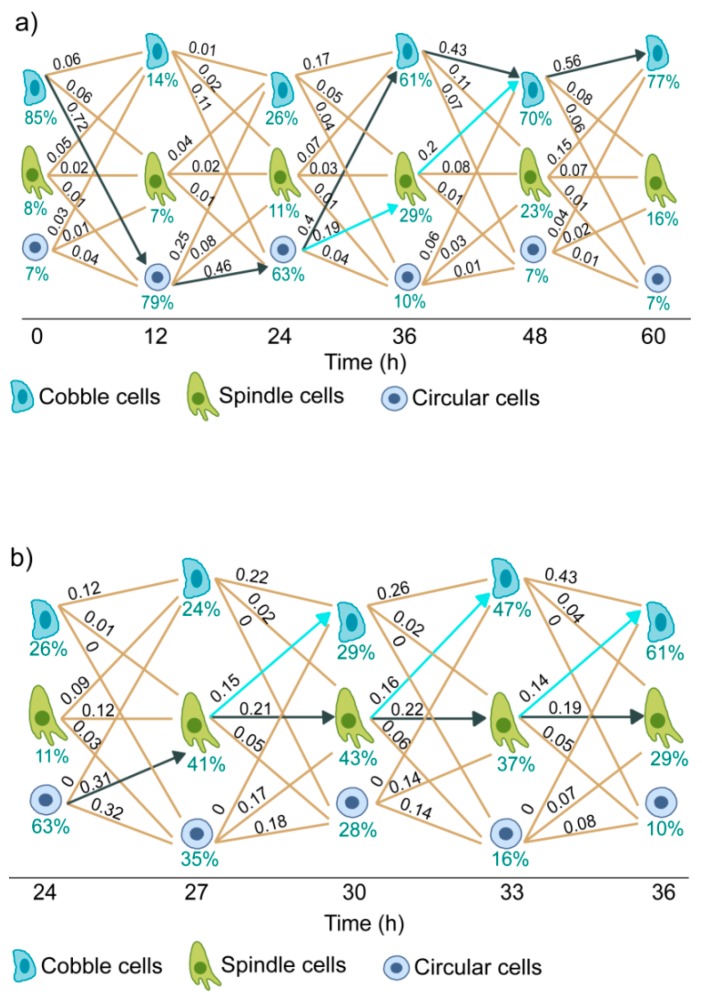
Cell state transition diagram of MDA-MB-468 cells treated with 10 ng/mL of EGF. Each line represents one state transition path. State transition parameters were estimated from the quantitative imaging data. Numerical values over the lines indicate the normalized flow of cells through those paths. Pointed black arrows show the dominant transition path and pointed blue arrows indicate the next dominant transition path. (**a**) State transition trajectories for observations at 12 h intervals till 60 h post-EGF treatment and (**b**) State transition trajectories for observations at 3 h intervals in the period of 24 to 36 h.

**Figure 5 jcm-08-00911-f005:**
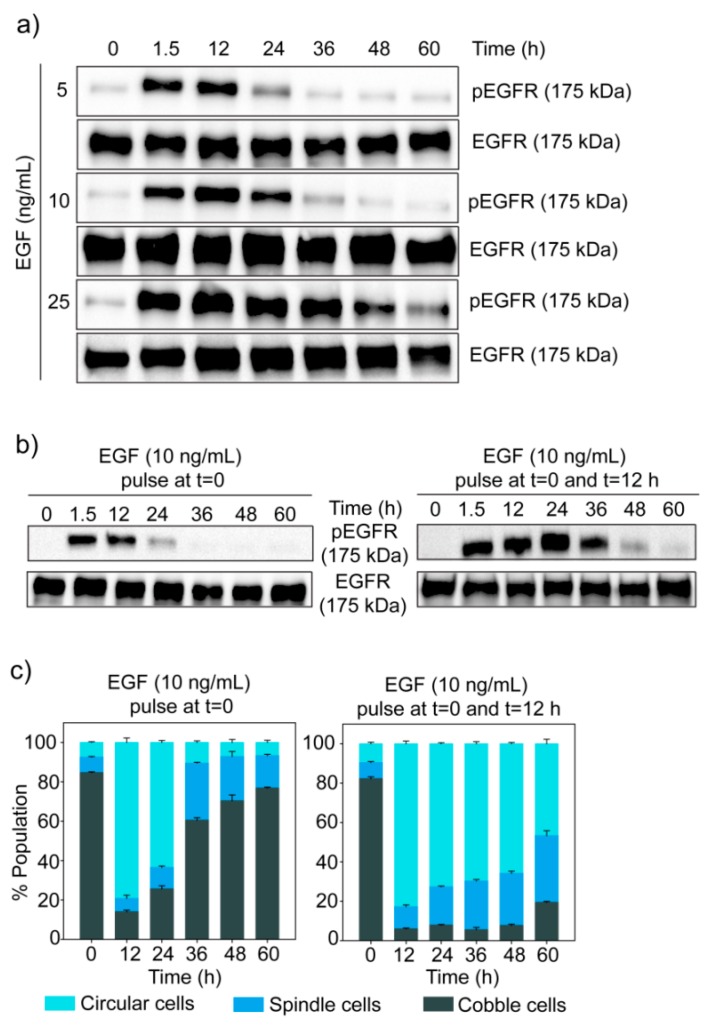
Temporal dynamics of phospho-EGFR in EGF treated MDA-MB-468 cells. (**a**) Cells were treated with different doses of EGF and phosphorylation of EGFR was measured at different time points by Western blotting. The experiment has been repeated three times and images of a representative experiment are shown. (**b**) MDA-MB-468 cells were given two pulses of EGF, and phospho-EGFR levels at different time points were measured by Western blotting. (**c**) Population distribution of MDA-MB-468 cells treated with two pulses of EGF. The graph represents data from quantitative image analysis. Each data point represents the mean of three independent experiments and error bars indicate standard deviation.

**Figure 6 jcm-08-00911-f006:**
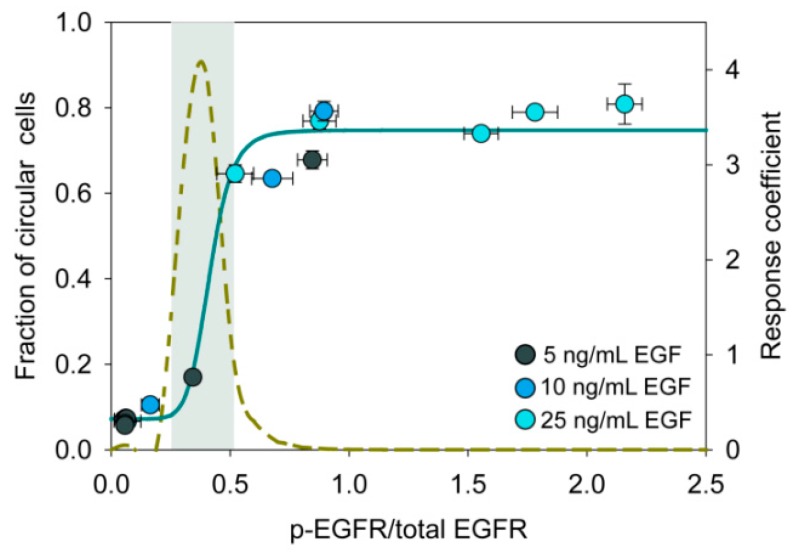
Ultrasensitive switch-like response during state transition. The plot shows the relation between the fraction of cells in circular state and phosphorylation of EGFR. Normalized level of phospho-EGFR was estimated by densitometry of Western blot images. Data were fitted to the Hill function (Hill coefficient *t* = 8.6). Ultrasensitive systems have a Hill coefficient greater than one. The dashed line represents the response coefficient [53]. The gray shaded region represents the ultrasensitive region where the response coefficient is greater than 1.

**Figure 7 jcm-08-00911-f007:**
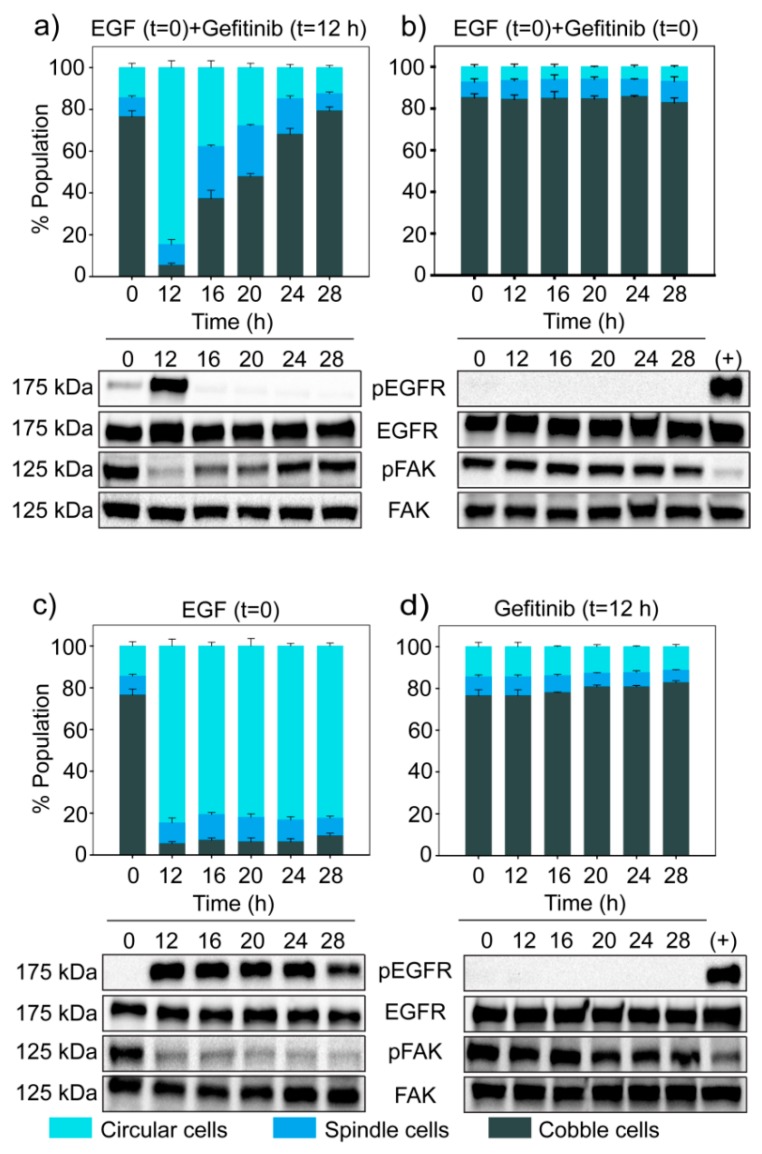
Blocking EGFR turns the ultrasensitive switch OFF. MDA-MB-468 cells were treated with different experimental conditions—(**a**) EGF (25 ng/mL) at *t =* 0 and Gefitinib (0.2 µM) at *t =* 12 h, (**b**) EGF (25 ng/mL) and Gefitinib (0.2 µM) together at *t =* 0, (**c**) only EGF (25 ng/mL) at *t =* 0, and (**d**) only Gefitinib (0.2 µM) at *t =* 12 h. For all the experimental conditions, EGFR, phospho-EGFR, FAK and phospho-FAK were measured by Western blotting and the corresponding population distribution was estimated through quantitative image analysis. EGF treated samples were used as positive control for phospho-EGFR in (b) and (d). The bar graph represents quantitative data from image analysis. Each data point represents the mean of three independent experiments and error bars indicate standard deviation.

**Figure 8 jcm-08-00911-f008:**
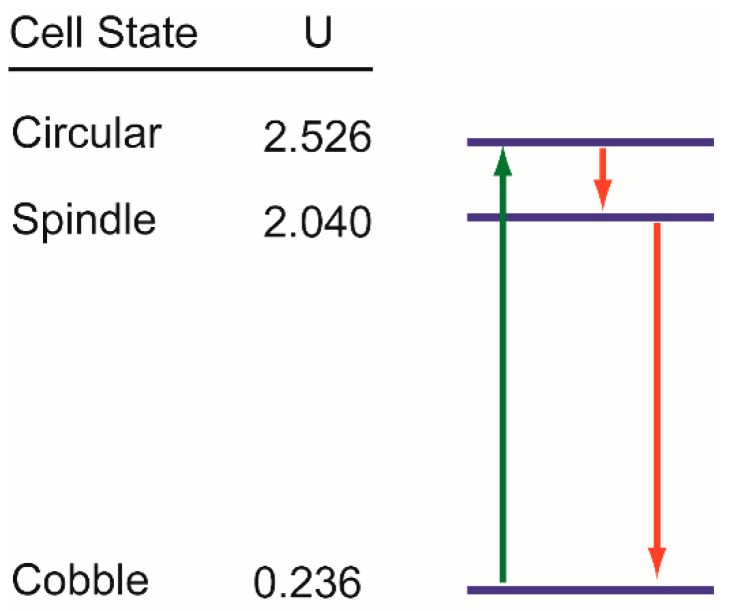
State diagram to understand state transition dynamics. Each morphological state corresponds to a specific discrete energy level in this diagram. These energy levels are shown by blue horizontal lines. Corresponding potentials (U) were calculated from the steady-state population distribution of cells in different states. State transitions are probabilistic and only the dominant state transition paths are shown here using green and red lines. All these state transitions were observed within 60 h of observations in cells treated with 10 ng/mL of EGF.

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
