# Peer review of "Morphological State Transition Dynamics in EGF-Induced Epithelial to Mesenchymal Transition"

_jcm, 2019, doi:10.3390/jcm8070911_

Round 1
Reviewer 1 Report
In the present study, Devaraj and Bose investigated the morphological state transitions during the epidermal growth factor (EGF)-induced EMT in MDA-MB-468 breast cancer cells. Though quantitative imaging and mathematical modeling, the authors analyzed the dose- and time-dependent effects of EGF on transition dynamics among three morphological states – cobble, spindle and circular. They showed that the dominant reversible state transition trajectory is cobble to circular to spindle to cobble. They also showed that there exists an ON/OFF switch involving phospho-EGFR that decides whether a cell will be in the circular state or not. Overall, the study has been well designed and performed.
the authors have to address the following comments.
1. What is the different molecular characteristics of the three distinct morphological states – cobble, spindle and circular?
2. Cells undergoing EMT can acquire hybrid epithelial/mesenchymal (E/M) phenotypes which combine epithelial and mesenchymal traits. What is the connection between the morphological states and epithelial, mesenchymal and hybrid E/M phenotypes as described in refs 7, 8 and 19?
3. Can the discrete-time population dynamics model (Fig. 4) capture the ultrasensitive switch-like response (Fig. 6)?
Author Response
Response to the comments of Reviewer 1:
We thank the reviewer for reviewing the manuscript and making the valuable comments. Here are our responses listed for each of the comments.
Comment 1. What is the different molecular characteristics of the three distinct morphological states – cobble, spindle, and circular?
Response: Change in shape of a cell involves several molecular processes, and it will be interesting to characterize cells of different shapes in terms of crucial molecules involved in cellular morphology. One can also characterize other important molecules like those involved in cell movement or can characterize in terms of EMT markers. We have not characterized/differentiated these three cell types in terms of molecular markers. We did immunofluorescence imaging experiments (Figure 1c) using key markers of EMT. However, as shown in that figure, although EGF-treatment induced the expected change in expression of these markers, the changes were not distinctly different in three cell types.
The focus of this manuscript is to study the phenotypic state transition and to connect it to the theory of state transition on a potential landscape. We have considered cell shape as a quantifiable phenotype that changes during EMT. The work primarily deals with the phenomenon rather than molecular processes. Though crucial, but we believe molecular characterization of these cells would be an independent work focused more on molecular processes behind the phenomenon. As mentioned in the discussion section of the manuscript, data of molecular markers along with phenotypic data (like cell morphology, migration) would be complementary to each other and would enhance our understanding of EMT further.
Comment 2. Cells undergoing EMT can acquire hybrid epithelial/mesenchymal (E/M) phenotypes which combine epithelial and mesenchymal traits. What is the connection between the morphological states and epithelial, mesenchymal and hybrid E/M phenotypes as described in refs 7, 8 and 19?
Response: In the above-said references, epithelial, mesenchymal, and hybrid phenotypes are classified based on the relative expression of specific molecular markers. These phenotypes can also be distinguished based on functional features. Epithelial cells are adherent and grow in clusters. Mesenchymal cells are less adherent and are migratory.
In our work, we have categorized cell phenotype based on morphology (cobble, spindle, and circular). In our experiments, we observed that the cobble cells grew in colonies while spindle and circular cells were scattered (supplementary figure S2). Also, we observed that cobble cells were non-migratory, while spindle and circular cells were migratory (supplementary figure S3). From these observations, we speculate that cobble cells are of epithelial phenotype, whereas spindle and circular cells are of mesenchymal phenotype.
Hybrid E/M phenotype are cells that express both epithelial and mesenchymal markers but have migratory potential. In our experimental system, when there was no EGF stimulation, cells were mostly in the epithelial state (cobble – non-migratory and exist in a colony). Upon EGF stimulation, most cells moved to mesenchymal state (circular – more migratory and scattered). During the decay of EGF signal, when cells started reverting from mesenchymal to epithelial state, we observed an increase in spindle cells (migratory and scattered). These spindle cells were intermediatory between circular state and cobble state. Based on these observations, one can speculate that most probably the spindle cells are of hybrid phenotype.
However, we believe mapping of categorization of cells based on a particular phenotypic measure (here morphology) with gene expression-based categorization of cells would be speculative right now. That is why we prefer to refrain from making any speculative comment on this issue in the manuscript.
Comment 3. Can the discrete-time population dynamics model (Fig. 4) capture the ultrasensitive switch-like response (Fig. 6)?
Response: The ultrasensitive behavior shown in Figure 6 explains the population dynamics observed in Figure 4.
In Figure 4, we have shown that there was a rapid and sharp increase in the circular cell population after EGF treatment. In fact, majority cells turned to the circular shape. Though Figure 6 shows such an increase in Circular cells at 12 hr, we have observed such changes at the very early time points. As shown in Figure 5a, the phosphorylation of EGFR is very high as early as 1.5 hr and remained very high at 12 hr. So the data of Figure 6 and 5 together explains why we observed such a rapid and sharp increase in circular cells post-EGF treatment.
Post 24 hr level of phospho-EGFR dropped (Figure 5a), and following the sharp switch-like behavior expected for an ultra-sensitive switch, the number of circular cells started dropping rapidly. However, this time, they did not return directly to cobble state but followed the path Circular à Spindle à Cobble.
So, data of Figure 4, 5, and 6 are in sync with each other and are explanatory. However, we have not included the ultrasensitive switch in the population dynamic model used for Figure 4. It should be noted that the sole purpose of the discrete-time population dynamic model used in the manuscript is to extract the most probable paths of cells state transition from data. We had discrete-time aggregate data of population distribution. The quantitative imaging data did not provide any information on the state change paths followed by cells. Therefore, we had the task to estimate the probable paths followed by the cells from the imaging data. That has been achieved by the mathematical analysis. The parameters of the fitted model allowed us to identify the paths followed by the cells during cell state transition.
The purpose of the model was not to explain a well-delineated population dynamics. That is why we have not included any mechanistic aspects (like an ultrasensitive switch) in the model. Inclusion of any mechanistic aspect in the model would have increased the number of parameters and would have made the parameter estimation less powerful.
Now we know that the dominant path followed by cells in this process is cobble à Circular à Spindle à Cobble and the ultrasensitive switch of pEGFR is a key regulator of state transition. Therefore, one can now create a mechanistic population dynamic model and explore the process further. We believe that should be an independent work.
Reviewer 2 Report
This paper is a very interesting paper and Authors discuss new original data. The Authors evaluated the morphology of a cell to define its phenotypic state and investigated the dynamics of morphological state transition during Epidermal Growth Factor (EGF)-induced EMT. Using quantitative imaging and mathematical modeling the Authors suggested that Epithelial to Mesenchymal Transition (EMT) is a multi-state process and the spindle and circular states are the migratory phenotypes. The phenotypic state transition during EGF-induced EMT in these cells is reversible, and depends upon the dose of EGF and level of phosphorylation of the EGF receptor (EGFR). The dominant reversible state transition trajectory in this system was cobble to circular to spindle to cobble. They stated that there exists an ultrasensitive on/off switch involving phospho-EGFR that decides the transition of cells in and out of the circular state.
However there are some points that the Authors should eventually clarify to the readers:
The Authors developed their investigation about the cancer cell phenotype evaluating only 2-dimensional cultures (dishes) but when they have to consider which cell phenotype was able to migrate they obviously reported their results considering the 3-dimensional cultures (Polycarbonate cell culture inserts with 125 8-micron pore size, Thermo Fisher Scientific). Cancer cell phenotype can change from 2-dimensional to 3-dimensional cultures because cancer cell phenotype is strongly influenced by substrate or microenvironment culture (see: Collagen Fiber Array of Peritumoral Stroma Influences Epithelial-to-Mesenchymal Transition and Invasive Potential of Mammary Cancer Cells. J. Clin. Med. 2019, 8, 213; doi:10.3390/jcm8020213). In this paper another breast cancer cell line (MCF 7) clearly showed a cobblestone phenotype (“Cobblestone cells were polygonal with cell-to-cell contact”) in 2-dimensional cultures, but the same cells exhibited a globular/spherical shape (“circular” for the Authors) when grew in 3-dimensional cultures (Millipore filter in Boyden chambers with attractant to favor cell migration). Did the microenvironment of 3-dimensional culture (with attractant) stimulate cell migration and transition phenotype from cobble to circular? The Authors should explain/comment in their paper the data reported by J. Clin. Med. 2019, 8, 213.
The Authors reported: “In the absence of EGF, very few cells migrated to the other side of the membrane, and they were circular. However, after EGF treatment, a large number of cells were observed on the other side of the membrane, and they were circular and spindle cells. Therefore, circular and spindle cell states are the migratory phenotypes.” It is known that even without EGF treatment both circular/globular and spindle shaped cells are able to migrate or invade microenvironment (collagen barriers). Can the Authors comment or clarify these apparently conflicting data?
Author Response
Response to the comments of Reviewer 2:
We thank the reviewer for reviewing the manuscript and making the valuable comments. Here are our responses listed for each of the comments.
Comment 1: The Authors developed their investigation about the cancer cell phenotype evaluating only 2-dimensional cultures (dishes) but when they have to consider which cell phenotype was able to migrate they obviously reported their results considering the 3-dimensional cultures (Polycarbonate cell culture inserts with 125 8-micron pore size, Thermo Fisher Scientific). Cancer cell phenotype can change from 2-dimensional to 3-dimensional cultures because cancer cell phenotype is strongly influenced by substrate or microenvironment culture (see: Collagen Fiber Array of Peritumoral Stroma Influences Epithelial-to-Mesenchymal Transition and Invasive Potential of Mammary Cancer Cells. J. Clin. Med. 2019, 8, 213; doi:10.3390/jcm8020213). In this paper another breast cancer cell line (MCF 7) clearly showed a cobblestone phenotype (“Cobblestone cells were polygonal with cell-to-cell contact”) in 2-dimensional cultures, but the same cells exhibited a globular/spherical shape (“circular” for the Authors) when grew in 3-dimensional cultures (Millipore filter in Boyden chambers with attractant to favor cell migration). Did the microenvironment of 3-dimensional culture (with attractant) stimulate cell migration and transition phenotype from cobble to circular? The Authors should explain/comment in their paper the data reported by J. Clin. Med. 2019, 8, 213.
Response: Thank you for pointing to the interesting paper by Franchi et al., J. Clin. Med. 2019, 8, 213. In this work, the authors have shown that MCF-7 cells are changing morphology when grown on Isopore Membrane Filter (Millipore). It is true that the extracellular matrix may affect the morphology of a cell. However, we did not observe any change in morphology of MDA-MB-468 Cells when cultured in transwell inserts in 24 well plates (Polycarbonate cell culture inserts with 8-micron pore size, Thermo Fisher Scientific).
It should be noted that all three types of cells, namely cobble, spindle, and circular, were present in untreated cells (No EGF) (Figure 3a). In untreated condition, ~13% cells were of spindle shape, and ~8% cells were circular. Treatment of EGF disturbed this proportion of three cell types and increased the number of circular and spindle cells (as shown in Figure 3c).
We have observed the migration of cells in the transwell experiment in both untreated and EGF-treated condition. In the absence of EGF, a minuscule number of cell migrated to the other side, and they were either circular or spindle. On the other hand, in the presence of EGF, the number of migrated cells was much higher, and those were again circular and spindle. This difference in cell migration was in accordance with the difference in the proportions of circular and spindle cells in untreated and EGF-treated condition.
In essence, we have not observed any special effect of the transwell membrane on the morphology of MDA-MB-468 cells, and EGF treatment increased the number of circular and spindle cells that are inherently migratory.
Considering the relevance of the paper by Franchi et al. (J. Clin. Med. 2019, 8, 213) we have discussed it in the revised manuscript (Page 7).
Comment 2: The Authors reported: “In the absence of EGF, very few cells migrated to the other side of the membrane, and they were circular. However, after EGF treatment, a large number of cells were observed on the other side of the membrane, and they were circular and spindle cells. Therefore, circular and spindle cell states are the migratory phenotypes.” It is known that even without EGF treatment both circular/globular and spindle shaped cells are able to migrate or invade microenvironment (collagen barriers). Can the Authors comment or clarify these apparently conflicting data?
Response: As explained in the response of the last comment by the reviewer, MDA-MB-468 cells were in three states - cobble, circular and spindle even in the absence of EGF (Untreated cells Figure 3, of Manuscript). In this condition, ~13% cells were of spindle shape, and ~8% cells were circular. Treatment of EGF disturbed this proportion of three cell types and increased the number of circular and spindle cells (as shown in Figure 3 in Manuscript).
The migration assay in the absence of EGF showed that spindle and circular cells migrated to the other side of the transwell (Supplementary Figure S3). This means that spindle and circular cells are inherently migratory. When the same experiment was performed with EGF-treated cells, more number of spindle and circular cells were observed on the other side of the transwell. That is expected as EGF treatment increased the number of spindle and circular cells.
From the migration assay (Supplementary Figure S3), we conclude that the circular and spindle cells are inherently migratory (in presence and absence of EGF). EGF stimulation only induces the formation of more number of circular and spindle cells.
We have reframed the text in Page 7 of the revised manuscript to clarify this issue.